# The Stress-Dependent Dynamics of *Saccharomyces cerevisiae* tRNA and rRNA Modification Profiles

**DOI:** 10.3390/genes12091344

**Published:** 2021-08-28

**Authors:** Yasemin Yoluç, Erik van de Logt, Stefanie Kellner-Kaiser

**Affiliations:** 1Department of Pharmaceutical Chemistry, Goethe University Frankfurt, 60438 Frankfurt, Germany; yoluc@pharmchem.uni-frankfurt.de; 2Department of Chemistry, Ludwig-Maximilians University Munich, 81377 Munich, Germany; vandelogt@genzentrum.lmu.de

**Keywords:** stress dependent RNA modification dynamics, absolute quantification of RNA modifications, isotope labeling, mass spectrometry, *Saccharomyces cerevisiae*

## Abstract

RNAs are key players in the cell, and to fulfil their functions, they are enzymatically modified. These modifications have been found to be dynamic and dependent on internal and external factors, such as stress. In this study we used nucleic acid isotope labeling coupled mass spectrometry (NAIL-MS) to address the question of which mechanisms allow the dynamic adaptation of RNA modifications during stress in the model organism *S. cerevisiae*. We found that both tRNA and rRNA transcription is stalled in yeast exposed to stressors such as H_2_O_2_, NaAsO_2_ or methyl methanesulfonate (MMS). From the absence of new transcripts, we concluded that most RNA modification profile changes observed to date are linked to changes happening on the pre-existing RNAs. We confirmed these changes, and we followed the fate of the pre-existing tRNAs and rRNAs during stress recovery. For MMS, we found previously described damage products in tRNA, and in addition, we found evidence for direct base methylation damage of 2′O-ribose methylated nucleosides in rRNA. While we found no evidence for increased RNA degradation after MMS exposure, we observed rapid loss of all methylation damages in all studied RNAs. With NAIL-MS we further established the modification speed in new tRNA and 18S and 25S rRNA from unstressed *S. cerevisiae*. During stress exposure, the placement of modifications was delayed overall. Only the tRNA modifications 1-methyladenosine and pseudouridine were incorporated as fast in stressed cells as in control cells. Similarly, 2′-O-methyladenosine in both 18S and 25S rRNA was unaffected by the stressor, but all other rRNA modifications were incorporated after a delay. In summary, we present mechanistic insights into stress-dependent RNA modification profiling in *S. cerevisiae* tRNA and rRNA.

## 1. Introduction

The central dogma of molecular biology states that DNA is the storage of the genetic code, which is transcribed into messenger RNA (mRNA) and translated into proteins with the help of transfer RNA (tRNA) and ribosomal RNA (rRNA). This fundamental life process is dominated by nucleic acids, which are composed of the canonical nucleosides adenosine, guanosine, cytosine and uridine (and thymidine in DNA). The sequence of these building blocks defines the genetic code of an organism. Additional chemical groups on these building blocks, commonly methylations, form a second layer of information on top of the code. In DNA, methylation was found to be dynamic. The addition or removal of a methylation on carbon C5 of cytosine can switch genes off or on [1,2]. Since this chemical code is additional information on top of the sequence, it is referred to as the epigenetic code. While epigenetics is an intensively studied area, the analogous process in RNA, termed epitranscriptomics, is far less studied [3]. This is mainly due to limited number of tools that can be used to study the dynamics of RNA modifications, and in addition, the complex process of finding biological consequences of RNA modifications. While DNA modifications must be removed by enzymatic or chemical processes to maintain the genetic sequence and its function, RNA has the option of simple degradation to disband unwanted RNA strands and the subsequent transcription of new RNA. This dynamic degradation and dilution, by new transcripts, constitutes a fundamental difficulty in the accurate assessment and quantification of RNA modifications. 

A fundamental study was presented by Chan et al. They provided insight into the changes to modifications in small RNA (<200 nts) as a cause of chemical stress exposure. This study coined the term “stress-dependent RNA modification reprogramming,” and there is clear evidence that RNA modifications are regulated by stress [4,5]. The methodological foundation of this and later studies was quantitative mass spectrometry, which allows one to assess changes in RNA modification abundance and compare, e.g., stressed samples with controls. A problem in the interpretation of the underlying data consists in the simultaneous analysis of RNA subspecies within the cell. For example, a higher modification density can be explained by additional modification events or by degradation of non-modified RNAs. A lower modification density is even more challenging to interpret. It can be caused by (a) enzymatic demodification processes, such as m^6^A for human mRNA [6,7] or ms^2^C in bacterial tRNA [8]; (b) by increased degradation of modified RNA; or c) by increased transcription of the RNA, without it being modified at all. As there are numerous RNAs within each cell and over two dozen abundant RNA modifications, it is very likely that a combination of all those processes is happening. To unravel the different mechanisms used for RNA modification adaptation, we developed stable isotope labeled pulse-chase studies in yeast [9], bacteria [10] and human cell culture [11]. Nucleic acid isotope labeling coupled mass spectrometry (NAIL-MS) relies on metabolic labeling of RNA to distinguish RNA modifications from different RNA subspecies, e.g., RNAs which existed during stress-exposure and RNAs which are transcribed during the stress recovery phase. For this purpose, we utilized isotope dilution mass spectrometry, which not only allows qualitative assessments of RNA modification changes, but furthermore, absolute quantification of RNA modifications [12].

In this work, we focused on the mechanisms which lead to the stress-dependent adaptation of tRNA and rRNA modifications during and up to 20 h after stress exposure in *S. cerevisisae*. We utilized stressors previously described by Chan et al., such as H_2_O_2_, MMS and NaAsO_2_. In addition, we studied the oxidant TBH and determined its impact on tRNA modifications. We applied our unique NAIL-MS technology to follow the fate of original RNAs exposed to stressors and how their modification profiles were impacted. For both tRNA and rRNA, we only observed minor changes. Only methyl-methanesulfonate which directly damages RNA [10], led to a substantial increase in methylated nucleosides. With a methylome discrimination assay, we proved direct methylation of the RNA, and we observed two undescribed RNA damage products which emerged from base methylation of 2′-O-ribose methylated nucleosides in rRNA. Furthermore, we closely looked at the original RNAs, but also new transcripts, during the time after stress exposure, when cells were striving to recover from the stressors. We found that cells exposed to stress barely showed signs of transcription. In addition, the speed of tRNA and rRNA modification during maturation slowed or even stalled after stress. Only some modifications, such as m^1^A and Ψ in tRNA and Am in rRNA, are more quickly incorporated into the new transcripts upon arsenite stress. These modifications are good candidates for future studies focusing on the role of RNA modifications in the stress response. 

## 2. Materials and Methods 

### 2.1. Chemicals and Reagents

All salts were obtained from Sigma Aldrich (Munich, Germany) at molecular biology grade, unless stated otherwise. Isotopically labeled compounds: ^15^N_2_- uracil (≥98% atom, Eurisotope), ^13^C_6_-glucose (≥99% atom, Eurisotope) and L-methionine-[^2^H_3_]-methyl (98 atom % D, Sigma-Aldrich). All solutions and buffers were made with ultrapure water (Milli-Q, Merck, Kenilworth, NJ, USA). Appendix A Appendix A shows all synthetic standards of modified nucleosides and their respective vendors.

### 2.2. Growth Media for S. cerevisiae

Yeast-nitrogen-base (YNB) (Carl Roth, Karlsruhe, Germany) minimal medium was prepared by mixing a 10× YNB stock solution supplemented with a mix of unlabeled aminoacids (final concentration in 1× growth medium: 0.02 g/L arginine, 0.02 g/L histidine, 0.06 g/L leucine, 0.03 g/L lysine, 0.05 g/L, phenylalanine, 0.4 g/L serine, 0.2 g/L threonine, 0.04 g/L tryptophane, 0.03 g/L tyrosine and 0.15 g/L valine). Depending on the desired stable isotope labeling, 0.02 g/L ^15^N_2_-uracil, 0.01 g/L ^13^C_6_-glucose, 0.72 g/L-methionine-[^2^H_3_]-methyl or their unlabeled isotopomers were used. 

### 2.3. Yeast Cultivation

A single colony of *S. cerevisiae* BY4741 was picked from a YPD-agar plate and used for inoculation of 5 mL YNB. The cells were grown at 30 °C at 250 rpm. The next day, the cell density was assessed by OD600 measurement (Eppendorf, Biophotometer plus) and the cell suspension diluted to OD 1. The cells were allowed to grow for 3 h to reach mid-log phase and the stressor was added. After 1 h of stress exposure the medium was exchanged by centrifugation (5 min, 3000× *g*, 24 °C). The resulting pellet was resuspended in fresh medium to initiate the recovery phase. Afterwards, 2 mL of cell suspension was harvested at set time points by centrifugation (5 min, 12,000× *g*, 4 °C). The RNA was isolated as described in Section 2.11.

### 2.4. LD_50_ Assay

For determination of LD_50_ values, yeast was cultivated as described in 2.3 and exposed to various concentrations of H_2_O_2_, MMS, NaAsO_2_, TBH or HOCl. After 1 h, 100 µL of each culture was diluted to 1/10^5^ with sterile water. From this dilution 70 µL was plated on a pre-warmed YPD agar plate. The YPD plates were incubated at 30 °C for 48 h and the colonies were counted.

### 2.5. Stress Study 

For assessment of tRNA and rRNA modification profiles under stress, yeast was cultivated as described in Section 2.3, split into a control and stress groups and exposed to the determined LD_50_ concentrations of H_2_O_2_ (2 mM), NaAsO_2_ (40 mM), HOCl (3 mM), MMS (12 mM) and TBH (10 mM). Then, 2 mL samples were drawn as indicated in the Figure 1, and the RNA was isolated and purified as described in Section 2.11, Section 2.12 and Section 2.13. 

### 2.6. Comparative NAIL-MS for Validation 

For validation of NAIL-MS conditions, *S. cerevisiae* BY4741 was grown overnight in unlabeled YNB medium or in ^13^C_6_-glucose/^15^N_2_-uracil labeled YNB medium, as described in Section 2.3. Before RNA isolation, 1 mL of labeled culture and 1 mL of unlabeled culture were mixed, and the RNA was immediately harvested (Section 2.11) before mass spectrometric analysis (Section 2.12 and Section 2.13, but without SILIS).

### 2.7. Pulse-Chase NAIL-MS Experiment

A 5 mL overnight culture of *S. cerevisiae* BY4741 was grown in ^13^C_6_-glucose and ^15^N_2_-uracil labeled YNB medium. The next day, the culture was diluted with ^13^C_6_-glucose and ^15^N_2_-uracil labeled YNB medium to OD 1 with a total volume of 32 mL. After 3 h of growth, the first sample was harvested. Afterwards the culture was split, and one half was exposed to the LD_50_ concentration of the respective stressor. After one hour of stress exposure, the next sample was harvested and afterwards the medium was exchanged by centrifugation (5 min, 3000× *g*, 24 °C). The resulting pellet was resuspended L-methionine-[^2^H_3_]-methyl labeled YNB medium. More samples were harvested (2 mL) at set timepoints after the initiation of the recovery phase. The RNA was extracted and purified as described in Section 2.11, Section 2.12 and Section 2.13.

### 2.8. Methylome Discrimination Assay 

L-Methionine-[^2^H_3_]-methyl labeled YNB medium was used, following the culturing method described in Section 2.3. After stress exposure, the stressor containing medium was removed and fresh L-methionine-[^2^H_3_]-methyl labeled YNB medium was used. 

### 2.9. Pulse-Chase NAIL-MS Experiment MMS Damage Repair 

To monitor MMS induced RNA damage repair, cells were grown as described in Section 2.3 using L-methionine-[^2^H_3_]-methyl labeled YNB medium in the overnight culture. The next day the culture was diluted with L-methionine-[^2^H_3_]-methyl labeled medium to OD 1, grown for 3 h and exposed to 12 mM MMS. After 1 h of exposure, the medium was exchanged by centrifugation (5 min, 3000× *g*, 24 °C), and the resulting pellet was resuspended in ^13^C_6_-glucose, ^15^N_2_-uracil and L-methionine-[^2^H_3_]-methyl labeled YNB medium. 

### 2.10. Knock-Out Screening

An overnight culture of each knock-out strain was grown in unlabeled YNB medium as described in Section 2.3, and 12 mM MMS was used as a stressor. After 1 h of stress exposure, samples were taken from the control and from the stress-exposed culture. Afterwards, the RNA was extracted and further processed for analysis.

### 2.11. RNA Extraction—Hot Phenol

Total RNA was isolated according to the hot-phenol extraction protocol of Collart et al. [13]. The extraction was followed by ethanol precipitation. Therefore, 0.1xV 3 M NH_4_OAc and 2.5xV 100% ice cold ethanol were added to the aqueous phase; the mixture was stored at −20 °C overnight. The next day, samples were centrifuged (40 min, 12,000× *g*, 4 °C), the supernatant was discarded and the reaction tube was rinsed with 200 µL of 70% ice-cold ethanol. After another step of centrifugation (10 min, 12,000× *g*, 4 °C), the supernatant was discarded, and the ethanol was air dried for 10 min. Afterwards, the total RNA was suspended in 50 µL H_2_O.

### 2.12. rRNA and tRNA Purification

18S and 25S rRNA and tRNA were purified by size exclusion chromatography (SEC) (AdvanceBio SEC 300 Å, 2.7 μm, 7.8 × 300 mm for tRNA combined with BioSEC 1000 Å, 2.7 μm, 7.8 × 300 mm for 18S and 25 S rRNA, Agilent Technologies) according to our published protocol [14]. After purification, the RNA was precipitated and dissolved in 30 µL H_2_O.

### 2.13. RNA Digestion for Mass Spectrometry

RNA (300–500 ng) in aqueous digestion mix (30 μL) was digested to single nucleosides by using 2 U alkaline phosphatase, 0.2 U phosphodiesterase I (VWR, Radnor, Pennsylvania, USA) and 2 U benzonase in Tris (pH 8, 5 mM) and MgCl_2_ (1 mM) containing buffer. Furthermore, 5 µg tetrahydrouridine (Merck, Darmstadt, Germany), 10 µM butylated hydroxytoluene and 1 µg pentostatin were added to avoid deamination and oxidation of the nucleosides. The mixture was incubated 2 h at 37 °C and then filtered through 96-well 10 kDa molecular-weight cut-off plates (AcroPrep Advance 350 10 K Omega, PALL Corporation, New York, NY, USA) at 3000× *g* and 4 °C for 30 min. Then, 1/10 Vol. of SILIS (stable isotope labeled internal standard) was added to each filtrate before analysis by QQQ mass spectrometry.

### 2.14. Preparation of rRNA and tRNA SILIS

*S. cerevisiae* BY4741 was grown in 5 mL of ^13^C, ^15^N Silantes rich growth medium (Silantes, Munich, Germany Product no.: 111601402) supplemented with 1% (*w*/*w*) ^13^C_6_-glucose. The culture was incubated overnight, and the next day it was diluted to OD 0.1 with fresh ^13^C, ^15^N Silantes rich growth medium supplemented with 1% (*w*/*w*) ^13^C_6_-glucose. The culture was incubated for another 2 days at 30 °C. The cells were harvested and RNA was extracted according to Collart et al. [13]. rRNA and tRNA were purified by size exclusion chromatography (SEC) (AdvanceBio SEC 300 Å, 2.7 μm, 7.8 × 300 mm, Agilent Technologies), as described in [14]. Subsequently, the RNA was hydrolyzed to single nucleosides, as described in Section 2.13. As an external standard 10 mM theophylline was added to a final concentration of 1 mM in the digestion solution. The resulting digest/theophylline mixture is referred to as 10 × SILIS, which was added to a final concentration of 1 x to samples and calibration solutions. The labeling efficiency was confirmed by high resolution mass spectrometry (HRMS). Spectra of precursor and product ions were recorded by a ThermoFinnigan LTQ Orbitrap XL operated in positive ionization mode after LC separation of ribonucleosides.

### 2.15. QQQ Mass Spectrometry

For quantitative mass spectrometry, an Agilent 1290 Infinity II equipped with a diode-array detector (DAD) combined with an Agilent Technologies G6470A Triple Quadrupole system and electrospray ionization (ESI-MS, Agilent Jetstream) was used. Operating parameters: positive-ion mode, skimmer voltage of 15 V, cell accelerator voltage of 5 V, N_2_ gas temperature of 230 °C and N_2_ gas flow of 6 L/min, sheath gas (N_2_) temperature of 400 °C with a flow of 12 L/min, capillary voltage of 2500 V, nozzle voltage of 0 V and nebulizer at 40 psi. The instrument was operated in dynamic MRM mode (multiple reaction monitoring, MRM). Mass transitions for all monitored analytes and their isotopologues are found in Appendix A. For separation a Core-Shell Technology column (Synergi, 2.5 μm Fusion-RP, 100 Å, 100 × 2 mm column, Phenomenex, Torrance, CA, USA) at 35 °C and a flow rate of 0.35 mL/min were used in combination with a binary mobile phase of 5 mM NH_4_OAc aqueous buffer A, brought to pH 5.6 with glacial acetic acid (65 μL), and an organic buffer B of pure acetonitrile (Roth, LC-MS grade, purity ≥.99.95). The gradient started at 100% solvent A for 1 min, followed by an increase to 10% over 3 min. From 4 to 7 min, solvent B was increased to 40% and was maintained for 1 min before returning to 100% solvent A and a 3 min re-equilibration period.

### 2.16. Calibration

For calibration, synthetic nucleosides were weighed and dissolved in water to a stock concentration of 1–10 mM. The calibration solutions ranged from 0.3 to 500 pmol for each canonical nucleoside and from 0.3 to 500 fmol for each modified nucleoside and were spiked with 1/10 volume of SILIS. The sample data were analyzed by MassHunter Quantitative Software from Agilent. The areas of the MRM signals were integrated for each modification and their isotopologues. The absolute amounts of the modifications were referenced to the absolute amounts of the respective canonical. In the case of the pulse-chase experiment, the different isotopomers were referenced to their respective labeled canonicals, so that original modifications were referenced to original canonicals and new modifications were referenced to new canonicals. 

### 2.17. Statistics

All experiments were performed at least three times (biological replicates) to allow student *t*-test analysis. The *p*-values of the Student’s *t*-test (unpaired, two-tailed, equal distribution) were calculated using Excel or Graphpad Prism.

## 3. Results

### 3.1. S. cerevisiae’s Total RNA Composition Is Changed by Chemical Stress Exposure

Intrigued by the concept of stress-dependent RNA modification reprogramming [4], we set out to study the reaction of *S. cerevisiae* on the transcriptome level in more detail. For this purpose, we wanted to use our established NAIL-MS methodology which is based on controlled stable isotope nutrient’s addition to minimal medium [9]. NAIL-MS relies on yeast nitrogen based medium (YNB), which differs largely from the commonly used yeast extract peptone dextrose medium (YPD). Thus, we first determined the 50% lethal dose of every stressor for a *S. cerevisisae* BY4741 culture grown in YNB medium (Appendix A). For this purpose, an overnight yeast culture was diluted in YNB medium and grown for 3 h until mid-log growth phase and stressed with either methyl-methanesulfonate (MMS) or one of the oxidants: hydrogen peroxide (H_2_O_2_), arsenite (NaAsO_2_), tert-butyl hydroperoxide (TBH) or hypochloric acid (HOCl). After one hour of stress exposure, the cells were pelleted and resuspended in fresh YNB medium for recovery. The growth curve for the non-exposed control cells shows the unaltered growth of the cells, whereas the cells exposed to MMS and H_2_O_2_ showed a delay in growth after stress exposure (Figure 1). The cells exposed to the oxidants NaAsO_2_, TBH and HOCl did not recover within 24 h and showed no growth within this timespan. We next extracted the total RNA from cells after one hour of exposure using either the commercial TRI reagent and glass bead approach or hot phenol [13]. The total RNA was loaded onto a size exclusion chromatography column of 1000 Å, and the eluting RNA was detected using UV absorption at 254 nm. As shown in Figure 1b,c, the TRI based method yielded mainly RNAs smaller than 200 nts. 18S and 25S rRNA were of low abundance and undefined size. In contrast, the hot-phenol method yielded high amounts for 18S, 25S and tRNA. Judging from the elution profile in Figure 1c, the integrity of rRNAs remained under MMS and H_2_O_2_ exposure, whereas all rRNA was lost in cells exposed to the oxidants NaAsO_2_, TBH and HOCl. Overall, the profile of total RNA was bizarre in these cells, and the fate of the rRNA is unclear. Therefore, the stressor HOCl was not further pursued for RNA modification analysis. Only NaAsO_2_ and TBH showed acceptable integrity of tRNA, and thus tRNA modification profiles can be analyzed. 

### 3.2. tRNA Modification Reprogramming in S. cerevisiae Is Stress Dependent; rRNA Modifications Are Unaltered

For RNA modification analysis we used our established stable isotope dilution LC-MS/MS protocol [12]. In the acute phase, 60 min after stress exposure, we found changes in tRNA modification density in dependence of the chemical used, as previously suggested by Chan et al. [4]. A comparison of the published data and our fold-change data is given in Figure 1d. The direct comparison revealed several differences between our and the published data. For H_2_O_2_, we found less tRNA modification reprogramming, while similar trends are found for MMS and NaAsO_2_ exposure. We have identified three major experimental differences which contributed to the observed differences: (1) The published experiments were performed in rich YPD growth medium, whereas we used minimal YNB medium. (2) Chan et al. used a column affinity-based protocol for purification of RNA smaller than 200 nts, whereas we used size exclusion chromatography for tRNA purification. (3) Our mass spectrometric data was acquired using stable isotope dilution, which allows absolute quantification of RNA modifications. Therefore, we are confident that our data reflect the changes in tRNA modification profiles accurately. With our study we confirm the findings by Chan et al. that tRNA modifications are reprogrammed in the acute moment of chemical stress exposure and that the changes are dependent on the chemical stressor. However, especially for H_2_O_2_ exposure, the observed changes were minimal and were not statistically significant. For MMS, we found substantial formation of 1-methyladenosine (m^1^A), 3-methylcytidine (m^3^C), 6-methyladenosine (m^6^A) and 7-methylguanosine (m^7^G), as recently described as RNA main damage products [10,15]. From the same experiments, we purified the 18S and 25S rRNA and subjected them to RNA modification quantification by LC-MS/MS. After 60 min of stress exposure, we found only minor changes in the natural epitranscriptomes of both rRNAs. This is in accordance with a recent study from the Novoa laboratory [5]. However, in rRNA from MMS exposed yeast, we found high numbers of the potential damage products m^1^A, m^7^G, m^3^C and m^6^A and a damage-methylated 2′-O-methyladenosine (m^x^Am).

### 3.3. MMS Directly Methylates tRNA and rRNA in S. cerevisiae

After MMS exposure, we detected high abundances of those RNA modifications, which are RNA damage products that were described in *E. coli* studies [10,15]. With the goal of elucidating the origins of these RNA modifications in *S. cerevisiae*, we envisioned a methylation discrimination assay which distinguishes enzymatic RNA methylation from direct methylation damage. S-Adenosylmethionine (SAM) is the natural methyl-donor for yeast RNA methyltransferases, and by feeding L-methionine-[^2^H_3_]-methyl enzymatic methylations, they receive a +3 mass increase. As shown in Figure 2a, optimal labeling of native RNA modifications m^7^G, m^1^A and m^3^C was achieved with 18 mM L-methionine-[^2^H_3_]-methyl. Cells were exposed to 12 mM MMS in the continuous presence of 18 mM L-methionine-[^2^H_3_]-methyl. After one hour of MMS exposure, tRNA, 18S and 25S rRNA were extracted, and mass spectrometry analysis revealed the ratio of enzymatically placed methylations (*m*/*z* +3) to damage-derived methylations (*m*/*z* ± 0). As shown in Figure 2b, up to 60% of all m^7^G marks were caused by direct methylation with MMS. To a lower extent, m^3^C, m^1^A and m^6^A were caused by direct methylation of canonical nucleosides in tRNA. For rRNA, we found the same damage products (Figure 2c,d), and in addition, two base-methylated 2′-O-methyladenosine species designated as m^x^Am. A comparison to our synthetic standards of m^1^Am and m^6^Am indicates that the early eluting damage product was m^1^Am and the later one was m^6^Am (Appendix A). Both were caused by direct base methylation of the highly abundant Am of both rRNAs during MMS exposure. With the power of our methylome discrimination assay, we could clearly identify the origin of the base methylation from MMS and the enzymatic origin of the ribose methylation (Figure 2e). Intrigued by this finding, we searched for a methylation damage product of Gm in rRNA, and we observed a clear signal of m^7^Gm in the MMS exposed yeast samples. Further identification through the comparison with a synthetic standard has not yet been possible (Appendix A). A detailed analysis of the observed absolute quantities is given Appendix A.

### 3.4. Modification Density in Existing tRNAs Rises upon S. cerevisiae Stress Exposure

Although the changes in tRNA modification density were small, we were curious to find out how they emerged mechanistically. With a pulse chase NAIL-MS experiment, we aimed to answer the questions: How does transcription change due to stress exposure? Are existing tRNAs degraded? Is it the original tRNAs which are modified or even demodified? When do new transcripts emerge, and how quickly are they modified? To answer these questions, we required a robust NAIL-MS method which allows accurate and precise analysis of mainly methylated nucleosides. In our previous method published in 2017 [9], we established the necessary medium for such an experiment; however, the stable isotope labeled internal standard (SILIS) was problematic. For methylated cytidine derivatives especially, the same *m*/*z* was found in the SILIS, along with the original modification of the tRNA. Therefore, a new SILIS, without *m*/*z* overlap, with the analytes had to be produced. For this purpose, we utilized a commercially available yeast medium which was enriched with carbon-13 and nitrogen-15 instead of carbon-12 and nitrogen-14 atoms. For *S. cerevisiae*, glucose is the carbon source and an ideal energy source, and by addition of 0.1 g/L of ^13^C_6_-glucose, we received well growing cultures and high numbers of fully ^13^C-labeled yeast cells. The total ^13^C- and ^15^N-labeled RNA was isolated, the rRNA and tRNA were purified and the SILIS was prepared following our established protocol [12]. A comparison of our new SILIS and the previous SILIS is found in Appendix A and our recently published protocols [14]. 

With the new SILIS in hand, we followed our published yeast NAIL-MS protocol (Figure 3a). Briefly summarized, yeast is grown overnight in YNB medium supplemented with ^15^N_2_-uracil and ^13^C_6_-glucose. The next day, cells were brought to OD 1 in the same medium, left for 3 h to enter mid-log phase and then exposed to the chemical stressor. After one hour of exposure, the stressor was removed by medium exchange. For the chase phase, medium with L-methionine-[^2^H_3_]-methyl was used. Due to the mass spectrometric detection, we followed the abundances of modified nucleosides in RNA existing during exposure to the chemical, determined the abundance of new canonical nucleosides forming after stress and determined the modification incoperation in new transcripts.

In the first step, we compared the number of new canonical nucleosides to the number of original canonical nucleosides, which is an indicator of cellular metabolism. After two hours of growth in the new but stable isotope labeled medium, 50% of all tRNAs contained new canonical nucleosides. In contrast, all stressed cells contained less than 10% of new canonical nucleosides, which indicates that transcription of tRNA is substantially repressed during the stress recovery phase (Figure 3b). Thus, all RNA modification density changes observed in Figure 1d must have been derived from from changed modification patterns in the original tRNAs. To further investigate the impacts of stress on the tRNA modification profiles, we determined the number of modifications per original tRNA and compared the quantities of stressed cells to the numbers found in the respective control cells. Under acute stress, we mainly observed higher numbers of tRNA modifications for H_2_O_2_, MMS and TBH stress and lower numbers for NaAsO_2_ stress (Figure 3c). These observations are in good agreement with our findings in Figure 1d. 

Regarding the overarching hypothesis of stress-dependent RNA modification reprogramming, our findings concerning transcription rates (Figure 3b) indicate a non-active adaptation scenario for increases in tRNA modification abundance. The increase in tRNA modification density might have been caused by a combination of (a) halted transcription in stressed cells, which left only existing transcripts as substrates for RNA writer enzymes, and thus higher modification numbers were observed; and (b) ongoing transcription and slow maturation in the control cells, and thus we found lower modification numbers in the control cells. Lower numbers of modifications, as observed for NaAsO_2_, might indicate active removal of modifications; however, given the large number of different modifications that are affected, a global effect on mature tRNA might be causative. In theory, mature and thus modified tRNAs might be targeted for degradation. Yet, we see no evidence of increased degradation of original tRNAs (Figure 3b). The trend of decreased tRNA modification density in arsenite-exposed *S. cerevisiae* remained visible even after two hours of recovery. Our methodology is currently not able to determine the biological mechanism behind this intriguing observation. A more detailed overview and absolute numbers of all native modifications per tRNA, 18S and 25S rRNA can be found in Appendix A.

One of the most striking findings of Figure 3c is the direct methylation of nucleobases, which appears to have been reverted during the recovery phase. We designed a pulse chase assay based on the methylome discrimination methodology to follow the fate of enzymatically placed modified nucleosides and damage-derived nucleosides (Figure 4a). On average, both rRNAs receive more than a dozen damage methylations during the experiment, and at least one methylation damage is found per tRNA. We found for both tRNA and rRNA, unexpectedly fast loss of these damages within one hour of recovery. The abundance of native methylations remained unchanged (Figure 4b and Appendix A). We tested several knockout strains of known nucleic acid damage repair enzymes for their potential involvement in the demethylation process in *S. cerevisiae*. However, except for met18, which showed a decent involvement in total RNA demethylation, no enzyme was found to be part of an active demethylation machinery (Appendix A and Appendix A). Loss of damaged RNAs through targeted degradation of the damaged RNA subpopulation is another valid hypothesis, but our data on original-to-new transcript ratios in Figure 3B do not favor this hypothesis. In summary, rRNA and tRNA receive substantial methylation damage through MMS exposure of *S. cerevisiae*, but we do not know by which mechanism the damaged nucleosides were lost within 60 min of recovery.

### 3.5. tRNA Modification Placement in New Transcripts Is Stress Dependent

With NAIL-MS, we have the unique opportunity to observe the speed of modifications in RNAs transcribed after a stress event. Figure 5a shows the early formation of highly abundant and less abundant modifications in total tRNA from *S. cerevisiae*. In accordance with our results from 2017 [9], we found immediately high amounts for modified nucleosides, such as pseudouridine (Ψ) and 2′-O-methylguanosine (Gm). Interestingly, other modifications of the anticodon-stem-loop mcm^5^(s^2^)U, t^6^A and i^6^A also appeared early on in total tRNA (Figure 5d and Appendix A). For Ψ, an immediate placement was also observed in human total tRNA, human tRNA^Phe^ [11] and yeast tRNA^Phe^ [16]. However, for Gm, which is incorporated fast in yeast tRNA, we found slow incorporation in human tRNA. Other modified nucleosides such as m^1^A, m^5^C, Cm and m^7^G showed similar incorporation speeds to human tRNAs [11]. 

For ribosomal RNAs, we found an immediate steady-state abundance of ribose methylated modifications, as expected (Figure 5b) [17]. In 18S rRNA, m^7^G and in 25S rRNA, m^3^U, m^5^C and m^1^A, were also placed early on, which is unsurprising, given their later inaccessibility to modification enzymes in the mature ribosome. Previous studies proposed rRNA methylation as a co-transcriptional process in the early phase of ribosome assembly [17,18]. To our great surprise, the isomerization of uridine to Ψ was substantially slower, and it took more than two hours to reach the final modification density in both 18S and 25S rRNA. This observation is in stark contrast to our findings in human rRNA, where we observed a fast pseudouridinylation within minutes [11]. Our most puzzling result was observed for Ψ in the latest 20 h timepoint. Its abundance exceeded the steady state level observed in the unlabeled experiment (Appendix A). We excluded a methodological bias, due to the biologically valid data received for tRNA. Thus, this observation deserves continued research to discover the underlying mechanism.

For most tRNA and rRNA modifications, we observed immediate or extremely fast incorporation within one hour of transcription (Figure 5a–c). From our NAIL-MS stress exposure studies in Figure 3b, we now know that transcription is differentially impacted by the chosen stressors. However, how about the subsequent RNA modification processes? To address this yet unsolved question, we analyzed the emergence of modified nucleosides within new transcripts in the recovery phase using the NAIL-MS experiment from Figure 3c. For new tRNAs, we observed delayed incorporation of most modified nucleosides. This was exemplarily shown for m^5^C and mcm^5^s^2^U in Figure 5d, but also for Cm, m^3^C, Gm, m^1^G, m^2^G, m^22^G and m^7^G. For most modified nucleosides, the incorporation delay was strongest for the oxidants TBH and NaAsO_2_, and H_2_O_2_ and MMS had more modest delays. For rRNA, a similar delay in modification speed was observed. Cm formation was especially slower under stress in comparison to unstressed cells (Figure 5e,f). In contrast, m^1^A and Ψ in tRNA and Am in both rRNAs were incorporated as fast or even faster after stress and to a higher degree compared to the unstressed controls. In yeast tRNA^Phe^, m^1^A and Ψ were recently shown to be the starting point during maturation, which indicates their important role in this tRNA’s modification network [16]. NaAsO_2_ appeared to have increased m^1^A and Ψ abundances in newly transcribed RNAs, which might indicate involvement of these modified nucleosides in the arsenite stress response. 

Oxidative stress has a substantial impact on the cell, and thiolated biomolecules suffer especially from exposure to reactive oxidant species. For example, bacterial DNA can be naturally thiolated at a non-bridging oxygen of the phosphodiester bond. During exposure to hypochloric acid, the sulfur can be replaced by oxygen, which either causes lethal strand breaks or a regular phosphodiester bond [19]. Our NAIL-MS data from NaAsO_2_ and TBH stressed cells showed a potentially similar effect on thiolated tRNA modifications. In eukaryotes, methylthiolation of adenine at position 2 has been reported, and the resulting modifications ms^2^i^6^A and ms^2^t^6^A are known to reside in the anticodon stem-loop of tRNAs [3]. As shown in Appendix A, the abundance of original t^6^A increased during TBH exposure. In addition, both i^6^A and t^6^A were substantially more abundant in new tRNA transcripts compared to tRNAs from unstressed controls (Appendix A). Our NAIL-MS data indicate that thiolated nucleosides might be either direct substrates to reactive oxygen species or are impacted by the disturbed sulfur homeostasis.

## 4. Discussion

RNA and its modifications have gained renewed interest, and we are at the start of understanding their dynamic nature and their underlying mechanisms. In this study, we followed the fate of the model organism *S. cerevisiae* during stress exposure and stress recovery. As previously described by the pioneers in the field [4], we observed a stress-dependent change in tRNA modification abundance. Thanks to our study, we now have the mechanistic insight to hypothesize on the mechanism of tRNA modification profiling. Our data suggest that primarily RNA transcription, and especially the rates of activity of RNA modification enzymes, were causative of the reported changes. For 18S and 25S rRNA, we observed no or only minimal changes in RNA modification profiles due to stress in *S. cerevisiae*. However, due to the complete hydrolysis of RNA to nucleosides, it is possible that site-specific effects on RNA modifications found at multiple positions were lost in our analysis. 

Exposure to H_2_O_2_, NaAsO_2_ and TBH all resulted in oxidative stress, following different mechanisms. While H_2_O_2_ generates hydroxyl radicals by Fenton chemistry [20,21,22], arsenite stress induces indirect oxidative stress, which results in downregulation of RNA synthesis [23] and the overproduction of reactive oxygen species. Exposure to TBH leads to the generation of butoxyl radicals via a Fenton-type reaction [24]. Our analysis of both total RNA integrity (Figure 1b,c) and RNA modification adaptation substantiates the differential cellular effects of the oxidants. We think it is noteworthy that the common RNA composition of *S. cerevisiae* (85% rRNA, 10% tRNA and 5% mRNA) was scrambled during and after the stress exposure, which must be taken into account if RNA modification densities are quantified. 

In the context of methylation stress, we have previously reported on RNA damage products in *E. coli* and human cells [10,11]. Until now, we have observed the damages of canonical nucleosides and recently of thiolated RNA modifications. In this work, we described the methylation damage found on 2′-O-ribose methylated nucleosides Am and Gm, which are highly abundant in rRNA. We suggest that m^7^Gm is a new type of RNA damage; however, m^7^Gm requires further structural validation by comparison with synthetic standards, which are currently not available. In addition, we observed both m^1^Am and m^6^Am. However, similarly to m^6^A, we are unsure about the mechanistic origin of m^6^Am. Recently, the Dedon laboratory presented convincing evidence that m^6^A might emerge from m^1^A through dimroth rearrangement due to the RNA hydrolysis protocol required for LC-MS analysis [25]. Thus, it is possible that m^6^Am is in fact a secondary damage product of m^1^Am which undergoes dimroth rearrangement. 

Regarding the increases and sudden disappearance of all tRNA and rRNA methylation damage products, we can only speculate. In *E. coli*, the repair of m^3^C takes two hours through AlkB. m^1^A takes with more than four hours, and m^7^G is not removed at all. The slow speed of repair in combination with a clear substrate specificity observed in *E. coli* makes us wonder how all methylation damage types are repaired within one hour in *S. cerevisiae.* Maybe, there are no demethylases involved, but a more global mechanism. Interestingly, we studied methylation damage in human RNA after MMS exposure, and to our surprise, we observed only minor quantities of RNA damage products which might argue towards instantaneous repair of the damage or a more sophisticated detoxification processes [11]. Thus, enzymatic involvement in the repair cannot be excluded. However, regarding the various enzyme knockouts screened, we did not find a satisfying candidate for an involvement in RNA repair through demethylation. Thus, the question remains: how do eukaryotic cells remove RNA methylation damage? We hope to uncover the biology behind this question in future studies.

## Figures and Tables

**Figure 1 genes-12-01344-f001:**
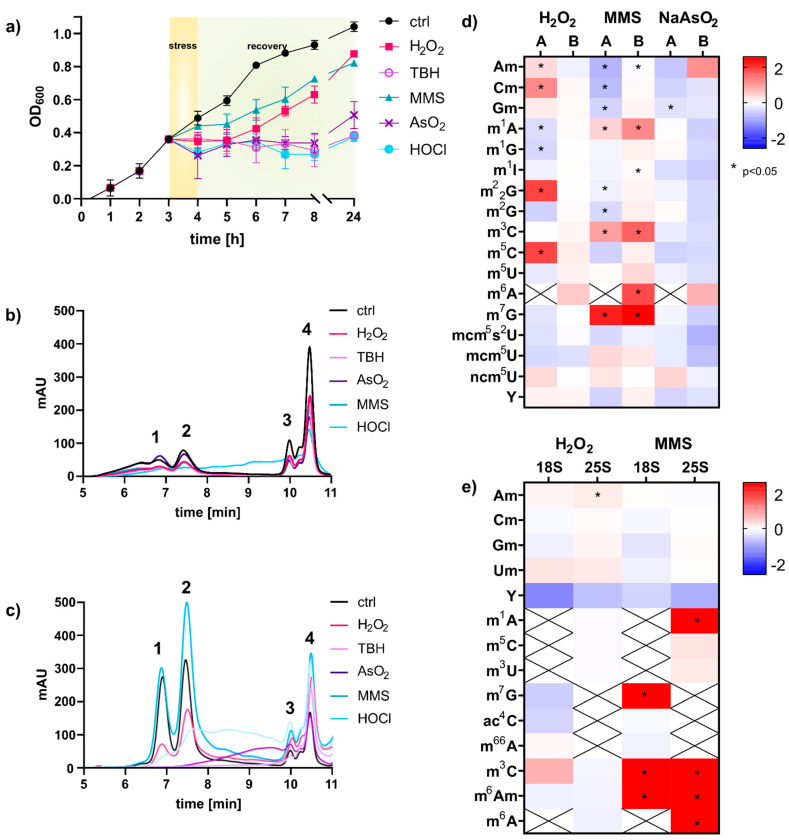
Analysis of total RNA and RNA modification changes upon chemical stress exposure of *S. cerevisiae* cultures. (**a**) Logarithmic growth curve in YNB medium using LD_50_ concentrations of stressors: H_2_O_2_ (2 mM), NaAsO_2_ (40 mM), HOCl (3 mM), MMS (12 mM), TBH (10 mM). (**b**) Total RNA of *S. cerevisiae* isolated with TRI reagent or hot phenol (**c**) assessed with size exclusion chromatography, 1: 25S, 2: 18S, 3: 5.8S rRNA, 4: tRNA (**d**) Fold changes in modification density in small RNA < 200 nts: (A) Chan et al. and (B) total tRNA from this work. An increase is shown in red and a decrease in blue. Student’s *t*-test results: *p* < 0.05 indicated with *. (**e**) Fold changes in modification density in 18S and 25S rRNA; Student’s *t*-test results: *p* < 0.05 indicated with *. All experiments were done in biological triplicates; error bars reflect standard deviations.

**Figure 2 genes-12-01344-f002:**
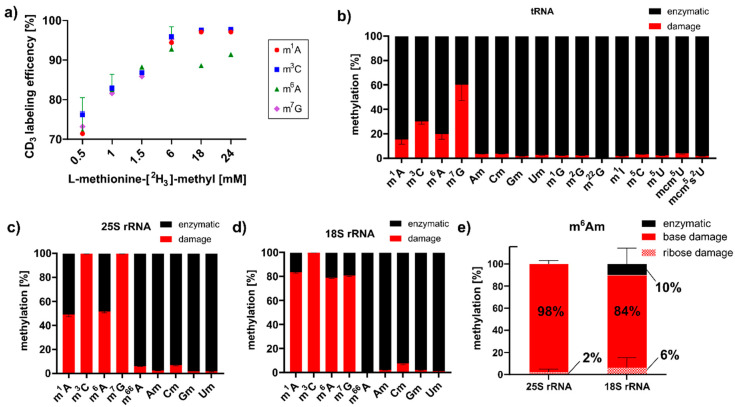
Methylome discrimination assay in *S. cerevisiae*. (**a**) Titration of optimal L-methionine-[^2^H_3_]-methyl supplementation to receive the highest abundance of CD_3_-labeled RNA modifications. (**b**–**d**) Ratios of damaged (red) to enzymatically (black) methylated nucleosides in *S. cerevisiae* tRNA (**b**), 25S rRNA (**c**) and 18S rRNA (**d**) after 60 min of MMS exposure. (**e**) The methylome discrimination assay reveals the origin of the methylation in the newly identified RNA damage product m^6^Am.

**Figure 3 genes-12-01344-f003:**
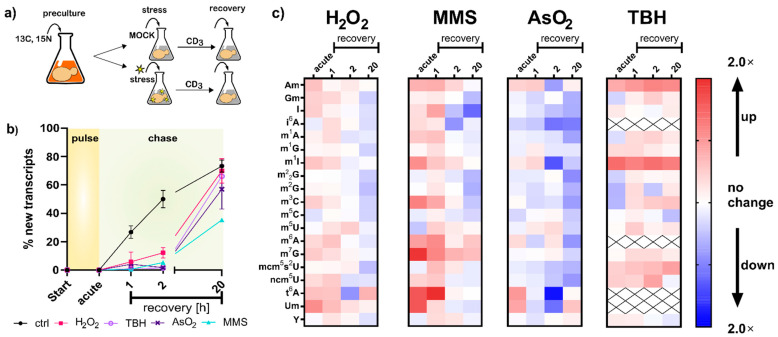
NAIL-MS pulse chase experiment with different stressors. (**a**) A concept sketch of the pulse-chase NAIL-MS assay. (**b**) Ratios of new and original tRNA transcripts displayed as percentages of new transcripts with the LD_50_ dose; data from *n* = 3 biological replicates; error bars reflect standard deviations. (**c**) Relative abundances of modified nucleosides in total tRNA compared to the time-matched control. The resulting fold changes in red indicate higher modification densities, and blue indicates lower abundances. Data are averages from three biological replicates.

**Figure 4 genes-12-01344-f004:**
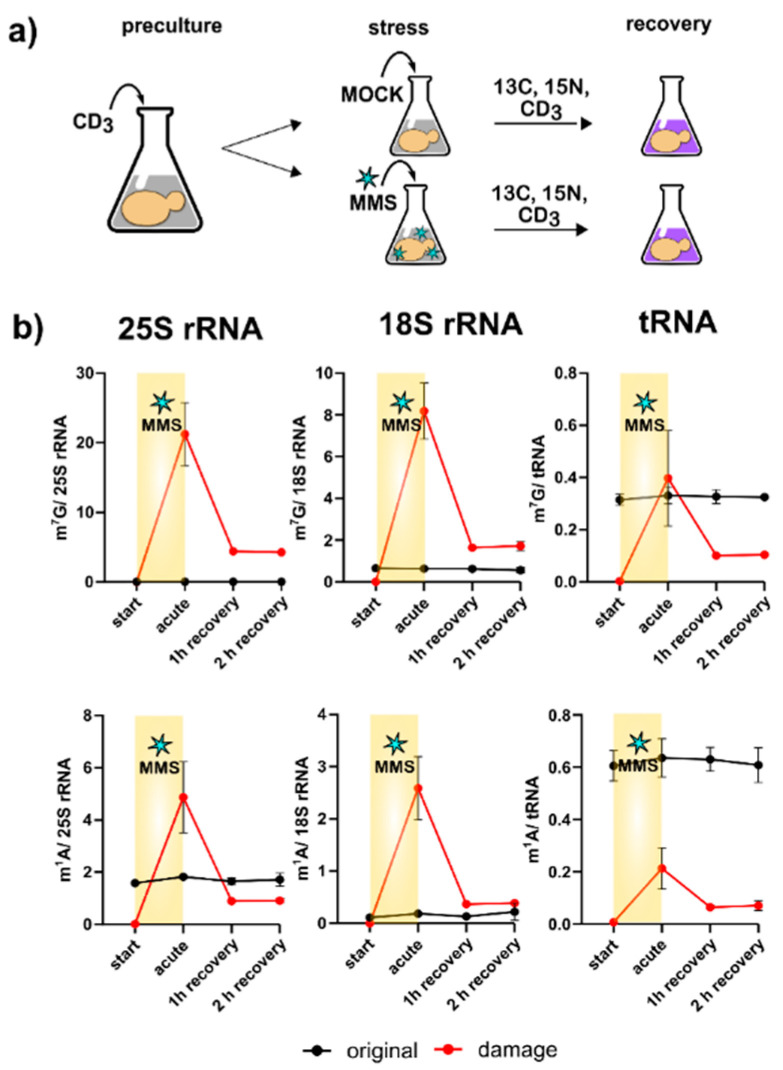
Methylome discrimination-based pulse chase experiment. (**a**) A concept sketch of the assay. Initially cells are grown in a medium containing L-methionine-[^2^H_3_]-methyl; after stress exposure the cells are cultivated in medium with L-methionine-[^2^H_3_]-methyl and ^15^N_2_-uracil, ^13^C_6_-glucose. (**b**) Numbers of m^7^G (upper panel) and m^1^A (lower panel) per respective RNA. Red line shows damage-induced methylation; black line shows enzymatic methylation. Data for tRNA from three biological replicates; error bars represent standard deviations. Data for 25S and 18S rRNA from two biological replicates; error bars represent standard deviations.

**Figure 5 genes-12-01344-f005:**
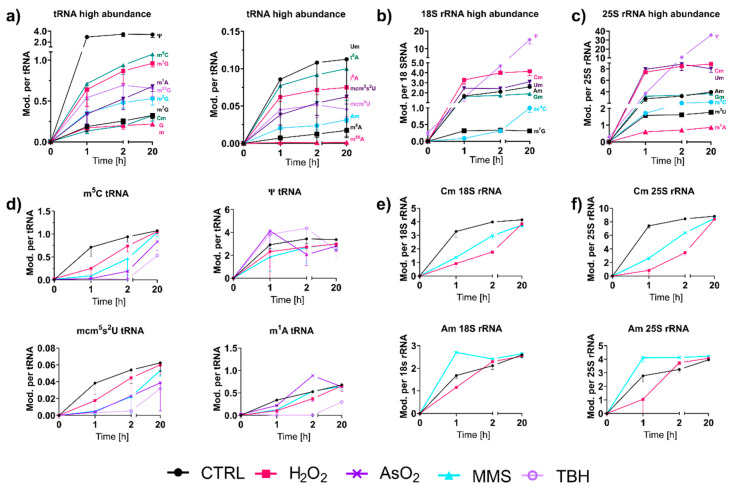
Incorporation of RNA modifications into new transcripts over time. Cells were initially grown in medium containing ^15^N_2_-uracil, ^13^C_6_-glucose; after stress exposure, the cells were cultivated in medium with L-methionine-[^2^H_3_]-methyl. (**a**) Abundances of modifications in new tRNA sorted by high and low abundances of modifications. (means of *n* = 3 and error bars reflect standard deviation). (**b**,**c**) Abundance of modifications in new 18S (**b**) and 25S (**c**) rRNA. (means of *n* = 2, and error bars reflect standard deviation). (**d**) A comparison of modification abundances in new total tRNA with dependence on stress. (means of *n* = 3, and error bars reflect standard deviations). (**e**,**f**) A comparison of modification abundances in new 18S (**e**) and 25S (**f**) rRNA in dependence on stress. (means of *n* = 2, and error bars reflect standard deviations).

## Data Availability

Not applicable.

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
