# Peer review of "The Stress-Dependent Dynamics of Saccharomyces cerevisiae tRNA and rRNA Modification Profiles"

_genes, 2021, doi:10.3390/genes12091344_

Round 1

Reviewer 1 Report

RNA modifications, their putative “dynamics” and the underlying functional consequences of these possible changes are emerging key questions enabling to better understand how RNA modifications-dependent (rapid) regulation may potentially alter biological functions. However, several technical challenges need to be overcome in order to obtain faithful RNA modifications map that integrate changes in RNA transcription and decay while keeping sufficient sensitivity to analyse dynamics within potential biologically relevant time scales.

In the current studies the authors have used a combination of semi-quantitative mass-spectrometry and metabolic labelling to analyse the consequences of various stress conditions on the fate of tRNA and rRNA modifications in the yeast S. cerevisiae.

The key conclusion of this study is that tRNA and rRNA modifications changes are limited under the stress conditions analysed, this being particularly true for the rRNA modifications. Moreover, the authors suggest that changes in tRNA modifications, in contrast to previous report, may be mainly explained by transcriptional effect/decay rather than RNA modification dynamics in the conditions tested.

General comments:

The manuscript is potentially interesting from a biological and methodological point of view; however, I must admit that the authors message is not always coming with strong clarity to the readers, and I had to read some parts multiple times to be sure of the authors’ real intention/message. Similarly, there are some methodological choices that needs to be better justified. There are also some points in the discussion that are in my opinion not sufficiently documented or supported. Finally, there are also a few typos/errors most of which could have been avoided by more rigorous scrutinization.

Specific comments:

1) The first introductory paragraph contains no references to existing literature!!!

2) Yeast culture/experimental design: according to the main text and method part, yeast cells were diluted in YNB to OD 1 before additional manipulations are done. The growth curve shown in Figure 1a are almost reaching OD 12!

I have honest doubt on the cultivation conditions, and I am wondering whether there is not a simple typo error here.

I believe the authors perform culture in regular flask whereby OD 1 in YNB would be rather a late-log phase/toward stationary after the remaining treatment/incubation.

Biologically speaking the growth phase studied is of importance!!

If I understood the intention of the authors, they aimed to stress the cells “upon entry into logarithmic growth phase” (l. 279), whereas as currently described the analysis would be more log-phase/stationary, where the dynamics range of RNA modifications might be more limited (?). The authors should clarify this point without which it is difficult to judge the underlying biology of the current study. Comparison with the Chan et al study is also done, in this study mid-log phase was used, this may be also (or not) a substantial difference here.

I am also surprised by the growth curve shown in Figure 1a, the wildtype (ctrl) does not look much to behave like a log phase growth as well. A semi-log scale would probably be more appropriate in this case.

Along those line I am wondering how a recovery of 20 hours (performed later in different experiments) can be meaningful in these ODs range. In any case the authors should provide more details on the growth conditions as it currently more puzzling than anything else and do not allow correct review of the biological context of the current results.

3) Figure 3b: Percentage of new transcripts. Assuming that the authors were performing the experiments in log-phase (see previous comment), I would expect that the percentage of new transcripts to reach almost 100% over time by mean of simple dilution effect: considering a doubling time of 2 hours, and a dilution effect of the “old transcript” of 50% per generation time… Please comment and/or provide sufficient description on this.

4) Figure 5: All the y-axis are labelled with tRNA regardless of the RNA type.

Experiments are performed over a 20 hours’ time course, assuming log-phase, RNA modifications should reach steady-state value and all the curves should show a saturation after several cell division. In some instance, like pseudouridylation of 18S or 25S rRNA or 18S acetylation it is not the case, even after 20 hours!!! These values should be close to the experimental steady state value provided in Suppl. data and are unlikely to go beyond these values (it reached almost 40 for the 25S, 18S also seems to be higher ?!, the axis is not precise enough to make a certain estimate, and there is no sign that these reactions are yet saturated). The timing is also making poor sense if cells are analysed in log-phase too. Showing that the experiments come to saturation/steady state is crucial to validate the accuracy and validity of these measurements. Considering doubling time and the kinetic (see also below) of ribosome biogenesis in yeast (in log phase) it does not make much sense, and I am not fully convinced about the accuracy of these measurements.

Dimethylation abbreviation m66A or m22A should be changed to m6,6A and m2,2G. Please check for other instance in the manuscript where these modifications are mentioned.

5) Figure 5 continue: Based on these analyses, the authors argue that they explored co-transcriptional RNA modifications. I politely but firmly disagree with this conclusion. The experiments provide information on the RNA modifications of newly synthesised RNA, at best. The time scale of the pulse-chase used will hardly provide co-transcriptional information. Beyond the fact that the growth phase is unclear, assuming log phase, most of the synthesised ribosome are already matured within the time scale used (see point 7) and in late log-phase early stationary, the authors need to demonstrate that they are analysing co-transcriptional RNA modifications and not post-transcriptional rRNA intermediates within this analysed time scale. The time scales that might be necessary to obtain such resolution and support any conclusion going beyond newly synthesized RNA (not to be confounded with co-transcriptional and/or nascent RNA) are not met and most probably out of reach of the current methodology (see e.g. PMID: 20347423, for a time frame estimation of rRNA synthesis in yeast growing in log-phase).

6) Discussion l. 535-537: “Regarding the increase and sudden disappearance of all tRNA and rRNA methylation damage products, we can only speculate. The timescale of less than one hour is too fast for enzymatic repair.” I am a bit puzzle by the time scale of enzymatic kinetic that the authors are expected for enzymatic repair, even though the authors are providing some examples (somehow the relevant primary citations are missing!?). A biologically meaningful repair should be naively faster than the sum of the transcriptional dilution effect/RNA decay happening in an activally dividing culture, and I might rather argue that an hour could be more than sufficient considering the doubling time of the biological system in log-phase.

Targeted degradation of non-functional/functionally impaired tRNA/rRNA is a mechanism that should be discussed and that could also easily explain this rather sudden decrease of damage modifications. Whereas in addition transcriptional dilution will apply, e.g.  after an hour of full transcription in log-phase growth will produce a large amount of fully functional new ribosomal subunit too (~ around 2000 * 60 new ribosome in yeast, see e.g.  PMID: 10542411).

7) Discussion: l. 545-547 “Although no studies on ribosome assembly timeframes have been performed in yeast, a study in E. coli following the 30 S subunit suggests assembly times of around 2 hours [15]” It is a little bit naïve from the authors ´side to believe that no kinetic studies or estimation have been performed in yeast, one of the most studied eukaryotic model organisms, and probably the best understood eukaryotic ribosome biogenesis pathway. Ribosome biogenesis in E. coli and yeast occur in minute scales e.g., PMID 4557192 or PMID: 20347423, the information provided, and the ensuing discussion is erroneous, moreover the cited E. coli study relate to in vitro assembly, which has kinetically nothing in common to the in vivo situation!!!

8) Discussion: l548-569 “In conclusion, we suggest that Ψs in rRNA have a major role in structure stabilization especially at the later and final steps of ribosome biogenesis”. I do not see any striking evidence coming from this study for that. See also comments above regarding my concerns on the pseudouridylation/”co-transcriptional analysis”.

9) Methods limitation: the method is not position-specific and for multiple modification of the same type per RNA molecules some information might be theoretically lost by averaging effect, e.g. if there are concomitant variations at different positions within the same RNA molecule, this may result to similar average modification per molecules, but different stoichiometry position-wise. Can this phenomenon be fully excluded in these experiments? The Results should be discussed accordingly.

Reviewer 2 Report

RNA modifications, especially tRNA modifications, play a role in maintenance of life activity. A technique for quantitatively deriving the amount of RNA modification in cells is required. The authors focused on the mechanisms which lead to stress dependent adaptation of tRNA and rRNA modifications and analyzed methylation with the improved NAILS-MS method. I understand the technique is important and is a field that is expected to develop further in the future. However, there are some points to be pointed out about this manuscript.

Comments;

Fig1 d) and e) : MMS concentration is 12 mM. But I could not find out the concentration of H2O2 and NaAsO2 in the analysis. Also, it is difficult to understand which state of stress exposure samples were being analyzed. It should be written on figure or figure legend.

Lane 418-421: Decreased numbers of modifications…. 

I’m sorry, but I couldn’t understand the meaning. Do you know that the effects of NaAsOs differ between mature tRNAs and unmodified or hypo-modified tRNA?

Lane 548-550: In conclusion, we suggest that puseudoUs in rRNA have….

Since it is not a suggestion that can be derived from the references and the results obtained by the authors, the sentence makes me feel something wrong.

Fig 5A: I couldn’t find the conditions for taking this data. It is easy to understand if it is written in the figure legend.

Minor comments;

Line 27: 2’O-  >> 2’-O-

Line 63-65: Need citations about what is written here.

Line 143: the “SEC” should be written as “size exclusion chromatography”, although written on line 200.

Line 203: "2" of "H2O" is not a subscript.

Line 227 and 245: Do “HRMS” and “MRM” mean “high-resolution mass spectra” and “multiple reaction monitoring”, respectively?  

Fig. 2: The symbol representing the attribution is confused with the data and should be moved.

Line 503-513: I think that this paragraph should be in the discussion part.

Line 531-534: About “the Dedon lab data” and m1A- > m6A conversion under alkaline condition should be cited.

Line 513, 554: Since “ROS” has only appeared twice, should be written “reactive oxygen species”.
